# Potential Causal Association between C-Reactive Protein Levels in Age-Related Macular Degeneration: A Two-Sample Mendelian Randomization Study

**DOI:** 10.3390/biomedicines12040807

**Published:** 2024-04-05

**Authors:** Byung Woo Yoon, Young Lee, Je Hyun Seo

**Affiliations:** 1Department of Internal Medicine, Chung-Ang University Gwangmyung Hospital, Gwangmyung 14353, Republic of Korea; bwyoon.md@gmail.com; 2College of Medicine, Chung-Ang University, Seoul 06974, Republic of Korea; 3Department of Applied Statistics, Chung-Ang University, Seoul 06974, Republic of Korea; lyou7688@gmail.com; 4Veterans Medical Research Institute, Veterans Health Service Medical Center, Seoul 05368, Republic of Korea

**Keywords:** age-related macular degeneration, Mendelian randomization, single-nucleotide polymorphisms

## Abstract

Researchers have proposed a possible correlation between age-related macular degeneration (AMD) and inflammation or C-reactive protein (CRP) levels. We investigated the potential causal relationship between CRP levels and AMD. Single-nucleotide polymorphisms (SNPs) associated with CRP exposure were selected as the instrumental variables (IVs) with significance (*p* < 5 × 10^−8^) from the genome-wide association study (GWAS) meta-analysis data of Biobank Japan and the UK Biobank. GWAS data for AMD were obtained from 11 International AMD Genomics Consortium studies. An evaluation of causal estimates, utilizing the inverse-variance-weighted (IVW), weighted-median, MR-Egger, MR-Pleiotropy-Residual-Sum, and Outlier tests, was conducted in a two-sample Mendelian randomization (MR) study. We observed significant causal associations between CRP levels and AMD (odds ratio [OR] = 1.13, 95% CI = [1.02–1.24], and *p* = 0.014 in IVW; OR = 1.18, 95% CI = [1.00–1.38], and *p* = 0.044 in weight median; OR = 1.31, 95% CI = [1.13–1.52], and *p* < 0.001 in MR–Egger). The causal relationship between CRP and AMD warrants further research to address the significance of inflammation as a risk factor for AMD.

## 1. Introduction

Age-related macular degeneration (AMD) is one of the leading causes of visual impairment in developed countries, particularly among individuals above sixty years of age [1,2,3]. The prevalence of AMD has been increasing in recent times, adding to the healthcare burden [4]. Although its pathogenesis is not understood, multiple factors such as genetic predisposition, lipids, smoking, aging, complement factors, and oxidative stress play a role in its pathophysiology [1,5,6,7,8]. A recent genetic study revealed that several genes, including complement factor H (CFH) and C3, may be involved in the development of AMD [1,7,8,9,10]. Therefore, we hypothesized that the risk of AMD is based on the genetic risk score and predicted that its prevalence may increase as life expectancy increases [11]. Nonetheless, deviations from these predictions suggest that additional risk factors are implicated in the development of AMD and that more environmental factors are involved, as there are discrepancies between the real prevalence and the prevalence predicted using the sum of risk and genetic scores [11]. Hence, the identification of causal risk factors for AMD may facilitate its early detection and prevention. 

There are two types of AMD: early and late. Early AMD is characterized by drusen and/or pigmentary changes, whereas late AMD can belong to one of two subtypes: geographic atrophy (GA or dry) and choroidal neovascularization (CNV or wet) [4,12,13]. Inflammation is involved in both types of AMD, as the primary trigger of age-related degenerative diseases is oxidative damage leading to inflammation [14,15]. This inflammation causes early drusen and changes in the retinal pigmentation, while advanced stages are characterized by choroidal neovascularization [16,17].

C-reactive protein (CRP) is a homopentameric acute-phase inflammatory protein that was first discovered in serum in 1930 by Tillet and Francis while researching the serum of patients in the acute stage of pneumococcal infection [18]. CRP is secreted from the liver owing to pro-inflammatory cytokine stimulation, typically interleukin-6 (IL-6) [19]. The primary inducer of the CPR gene is IL-6; however, this in itself is not sufficient to activate the CRP gene [20,21]. IL-6 has a crucial role in triggering the acute-phase response and defending against infections in the liver [22]. IL-6 expression as also induced by the cytokine interleukin-1 (IL-1), which increases during inflammation. IL-1 is also induced in the liver and other tissues and has decoy receptors that inhibit inflammatory signaling, such as interleukin-1 receptor antagonist protein (IL-1RN) [23].

CRP is a biomarker of inflammation. However, CRP is not only a result of inflammation but also has both pro-inflammatory and anti-inflammatory roles, according to the form in which it is present: pentameric native CRP (nCRP) or monomeric CRP (mCRP). nCRP exhibits an anti-inflammatory pathway by activating the classical complement pathway, induces phagocytosis, promotes apoptosis, and inhibits nitric oxide (NO), whereas mCRP promotes chemotaxis and the recruitment of circulating leukocytes to sites of inflammation, delays apoptosis, and generates NO [19]. Thus, most of the research regarding CRP is related to diseases associated with NO release, such as cardiovascular disease, stroke, and diabetes, and those associated with complement pathways, such as bacterial infections [24,25,26].

Recently, elevated CRP levels have been shown to mediate cognitive impairment [27]. As the retina is regarded as an extension of the brain, inflammation owing to CRP and its cytokines can also affect the retina [28]. Furthermore, the dissociation of nCRP into mCRP has been shown to play a potentially crucial pathogenic role in patients with the high-risk CFH single-nucleotide polymorphism (SNP) (Y402H) [29]. This SNP involving the CHF gene is considered one of the most significant genetic risk factors for AMD development [30].

Despite these results, there has been conflicting evidence from observational research about the relationship between circulating CRP levels and AMD risk [31,32,33,34,35,36,37]. Several investigations have demonstrated a positive association between elevated CRP levels and increased risk of AMD [31,32,33,34,35]. Despite this, two investigations revealed no correlation between increased CRP levels and an increased probability of AMD [36,37]. Furthermore, four studies showed that there is no association between CRP gene expression and risk of AMD [38,39,40,41]. Despite these findings, a recent study using Mendelian randomization (MR) showed that there is a positive correlation between serum CRP and AMD and that there is strong genetic evidence that higher CRP levels increase the risk for all forms of AMD [42].

Genetic variants linked to possible exposures are used as instrumental variables (IVs) in molecular epidemiology (MR) to assess their causal implications on disease outcomes [43,44]. Prior studies using MR have suggested variable evidence of an association between CRP and AMD (odds ratio [OR] = 1.26; 95% CI, 1.16–1.37) in people with European ancestry [42]. In addition, a study on the two-sample MR analysis methodology using large cohorts, such as the UK Biobank (UKB), reported that the bias of MR-Egger did not affect the inverse-variance-weighted (IVW) and weighted medians [45]. Furthermore, the choice of IVs for CRP may affect the MR analysis results, and large datasets combining the meta-analyses of Biobank Japan (BBJ) and UKB are expected to generate more substantial results. To this end, we investigated the causal effects of CRP levels on AMD via two-sample MR using summary statistics from the BBJ and UKB meta-analyses as the exposures and the International AMD Genomics Consortium (IAMDGC) summary statistics as the outcomes [46,47].

## 2. Materials and Methods

### 2.1. Study Design

The Veterans Health Service Medical Center’s Institutional Review Board approved the study protocol (IRB No. 2023-03-005) and waived the mandate for informed consent due to the study’s retrospective design. This research was carried out in accordance with the Declaration of Helsinki.

### 2.2. Data Sources 

Figure 1 depicts a diagrammatic representation of the analytical study design. To examine the causative impact of CRP on the likelihood of developing AMD, we selected the dataset as follows: (1) exposure data were derived from the summarized statistics of the genome-wide-association-study (GWAS)-based meta-analysis conducted on the BBJ and UKB (*n* = 436,491 for CRP) (Table 1) [46]; (2) outcome data were derived from the summary statistics of the 11 sources of IAMDGC GWAS data (*n* = 105,248; [14,034 cases vs. 91,214 controls]) [47]. Table 1 lists the datasets used for the summary statistics.

### 2.3. Selection of the Genetic Instrumental Variables

SNPs associated with CRP exposure at the GWAS threshold (*p* < 5 × 10^−8^) were used as IVs. To ensure that each IV was independent of the others, these SNPs were pruned based on linkage disequilibrium LD (*r*^2^ = 0.001, clumping distance = 10,000 kb). To calculate the LD for the clumping process, we used the 1000 Genomes Phase III Dataset (European population) as the reference panel. The *F*-value was determined using the formula,
*F* = *R*^2^(*n* − 2)/(1 − *R*^2^),(1)
where *n* is the sample size and *R*^2^ is the proportion of exposure variance by genetic variance [48].

*F*-values greater than 10 indicate that there is no evidence of weak instrument bias [49].

### 2.4. Mendelian Randomization 

The following assumptions were made for IVs while conducting the MR analysis: (1) the IVs need to show a significant correlation with the exposure, (2) they should be independent of the factors that might impact the exposure–outcome relationship, and (3) they should only influence the outcomes via exposure, indicating that there is no directional horizontal pleiotropy effect. We employed inverse-variance-weighted (IVW) MR with multiplicative random effects as the primary method [49,50,51]. We also employed the weighted median, MR–Egger (with or without adjustment via the Simulation Extrapolation [SIMEX] method) regression, and the MR pleiotropy sum of residuals and outliers (MR–PRESSO) [52,53,54,55]. The IVW method is most effective when all genetic variations satisfy the three assumptions for IVs [56]. The IVW estimation could be skewed if one or more of the variants are invalid [52]. Nonetheless, even in the event where 50% of the instrument variables are unreliable, the weighted-median approach provides accurate estimates of causality [52]. The MR–Egger technique allows the estimation of appropriate causal effects even when pleiotropic effects are present, permitting a nonzero intercept that demonstrates the average horizontal pleiotropic effects [53]. MR-Egger with SIMEX can be utilized to rectify the bias when the assumption of ‘no measurement error’ is violated [54]. The MR–PRESSO test adjusts the results of the IVW analysis for horizontal pleiotropy by identifying and deleting the outliers [55]. The heterogeneity for IVW and MR–Egger was evaluated using Cochran’s Q and Rücker’s Q statistics, respectively [50,57]. Directional horizontal pleiotropy was assessed using the MR–PRESSO Global Test. Therefore, the results were interpreted using an appropriate MR analysis method [58]. The *p*-values < 0.05 for Cochran’s Q statistic, Rücker’s Q′ statistic, and MR–PRESSO global test indicated possible pleiotropy in the genetic variations. The TwoSampleMR and simex packages (R version 3.6.3, R Core Team, Vienna, Austria) were used for all of the analyses.

### 2.5. Network Analysis of Selected SNP IVs 

Network analyses were performed using the selected IVs in Appendix A via the STRING database (https://string-db.org, accessed on 3 October 2023) [59]. We selected an *F*-value of over 100 for the analysis and used the full-string network.

## 3. Results

### 3.1. Genetic Instrumental Variables

We identified 204 IVs that had a significant association with CRP (*p* < 5.0 × 10^−8^) (Table 2). The *F*-statistics for CRP (113.47) used for MR were >10, exhibiting a minimal probability of weak instrument bias (Table 2 and Appendix A). Detailed information on the IVs is provided in Appendix A. 

### 3.2. Heterogeneity and Horizontal Pleiotropy of Instrumental Variables

The Cochran’s Q test for IVW demonstrated that the IVs for CRP (*p* = 0.042) were heterogeneous (Table 2). Therefore, we used a random-effects IVW approach. The Rücker’s Q test within the MR–Egger regression model approached the threshold of significance (*p* = 0.074), suggesting a potential chance of heterogeneity between the IVs. The MR–Egger regression intercepts also revealed a pleiotropic effect before and after the SIMEX adjustment (*p* = 0.012). The MR–PRESSO global test also indicated the presence of horizontal pleiotropic effects (*p* = 0.032). Hence, the MR–PRESSO/MR–Egger results were considered the primary outcomes based on prior research and were supported by the IVW results [58]. Although MR-PRESSO was recommended, no outliers were identified.

### 3.3. MR for Analyzing the Causal Association between CRP and AMD

CRP demonstrated a significant causal association with AMD based on analyses using the IVW method (MR OR = 1.13; 95% confidence interval (CI), 1.02–1.24; *p* = 0.014), weighted-median method (MR OR = 1.18; 95% CI, 1.00–1.38; *p* = 0.044), and MR–Egger (MR OR = 1.31; 95% CI, 1.13–1.52; *p* < 0.001) (Figure 2). The following scatter plot shows the genetic relationships between each SNP and AMD and CRP (Figure 3).

### 3.4. Network Analysis of SNPs between CRP and AMD

CRP was observed in the middle of the network (Figure 4), where CRP is associated with *NLRP3*, *APCS*, *IL1RN*, *IL6R*, *IL1F1*, and *SERPINA*. *SERPINA* is associated with *APOC2* and *HNF4A*, whereas *APOC2* is also associated with *NECTIN2* and *TOMM40* and *HNF4A* is associated with *ARNTL*, *HNF1B*, and *GCKR*. The PPI network showed 13 genes interacting with each other, with the PPI enrichment score *p*-value being 0.000323. IVs that influence AMD only through CRP and are associated with each other are shown in Table 3, with an *F*-value of over 100. F values over 10 signify that these genes are not associated directly with AMD but through CRP.

## 4. Discussion

Our research highlighted a possible causal association between elevated CRP levels and increased risk of AMD, where the MR–Egger OR was 1.31 [95% CI: 1.13–1.52; *p* < 0.001]. This is consistent with the reports of Han et al., where the OR of AMD owing to elevated serum CRP from MR–Egger (SIMEX) was 1.36 [95% CI: 1.21–1.55, *p* < 9.9 × 10^−7^] [42]. However, several differences were observed in this study. First, for exposure data SNP evaluation, we used a GWAS-based meta-analysis of BBJ and UKB instead of UKB only. Therefore, we used 204 IVs, whereas Han et al. used 44 SNPs [42]. Second, we used the summary statistics of 11 sources of IAMDGC GWAS data for the outcome data, consisting of SNPs from European descendants. Third, we performed a network analysis among IV SNPs with F values over 100. We selected 14 IV SNPs that had associations with CRP and tried to analyze the significance of how CRPs affected the retina through the liver–brain–retina axis. 

AMD is widely recognized as the primary contributor to vision loss in the aged population [1,2,3]. The major risk factors include aging, environmental factors, and associations with genetic variants. Genes associated with AMD susceptibility encode complement factors, including CFH, factor B, and the complement components C2 and C3 [60,61]. Along with the presence of complement and other inflammatory mediators in drusen, the investigation of ocular drusen via proteomic and histochemical methods has revealed the presence of inflammatory proteins and complement components responsible for local inflammation. These include C5, C9, CRP, amyloid A, fibrinogen, and vitronectin [62,63,64]. Furthermore, systemic immune activation may accelerate these processes. In relation to this, there is a positive correlation between elevated levels of circulating CRP and an augmented susceptibility to AMD [31,32,33,34,35]. The complement system is an important component of the innate immune response. The culmination of complement system activation results in the generation of a membrane–assault complex that has the capacity to induce cell lysis. The accumulation of membrane–assault complexes has been found to be higher in both the aging population and in those diagnosed with AMD than in control groups of a similar age. Moreover, many disorders associated with complement system activation have been independently associated with AMD [65]. Additionally, several genes involved in the complement pathway have been identified as significant contributing factors to AMD development [66,67].

An initial investigation into the correlation between increased CRP levels and the advancement of AMD was conducted by Seddon et al. [68]. As the disease progressed, a rise in circulating CRP was observed. The study observed that those grouped into low-, medium-, and high-risk categories for AMD had serum CRP amounts below 0.5, between 0.5 and 10.0, and beyond 10.0 mg/L, respectively [31,32,69]. Nevertheless, this relationship has not yet been confirmed universally [36]. A recent study conducted by the Seddon group demonstrated a correlation between elevated levels of circulating CRP and increased susceptibility to AMD, irrespective of the CFH genotype [68]. In addition, Bhutto et al. documented the specific localization patterns of FH and CRP in the aging ocular system [70]. The authors observed a significant correlation between CRP and fibroblast growth factor (FGF) levels in the macular tissue of patients with advanced AMD compared with control individuals of the same age group. In individuals with AMD, the presence of CRP labeling was shown to be upregulated in the Bruch’s membrane, drusen, and choroidal vessel walls, while the labeling of FH was observed to be reduced in comparison with the control group. The findings of this investigation show evidence that patients with AMD have an elevated inflammatory milieu in the macula and a reduced ability to control complement activation.

Chronic inflammation is characterized by persistent tissue injury and concurrent efforts to repair it, resulting in tissue remodeling and impaired function [71]. It serves as the prevailing pathological foundation for age-related ailments, including cardiovascular diseases, diabetes, cancer, Alzheimer’s disease, and AMD. Numerous physiological alterations occur during aging that actively contribute to the onset and progression of inflammatory responses [72]. Anti-inflammatory cytokines and acute-phase reactants increase with aging, which makes the immune system of the elderly unique due to a basic systemic inflammatory condition. 

The starting point for this study was a previous publication by Shin et al., in which the risk allele frequencies of AMD were used to calculate a polygenetic risk score to predict AMD between ethnicities [11]. Although this model fits well with the early and late AMD models for the <45-year age group, it does not fit well for ages above 65 years, as shown in the correlation plot of late AMR prevalence and genetic risk factors of Shin et al. [11]. We assume that environmental factors, such as CRP, may have played a role in explaining this gap, whereas the polygenetic risk score for AMD may not completely explain the discrepancy between the theoretical and real-world data [5,11]. This is congruent with a review by Feng et al., who concluded that CRP is more involved in late AMD than in early AMD, as patients with early AMD showed no difference in CRP levels [73]. Further research is needed to evaluate the role of the level and duration of CRP exposure in late AMD. 

Some considerations can be made based on the network analysis in Figure 4 of the resulting SNPs as IV. There are 13 genes associated with CRP, as in Table 3. Seven genes are anti-inflammatory when they increase in expression: *SERPINA1* (Serpin Family A Member 1), *HNF4A* (Hepatocyte Nuclear Factor 4 Alpha), *ARNTL* (aryl hydrocarbon receptor nuclear translocator-like protein 1), *HNF1B* (HNF1 Homeobox B), *GCKR* (Glucokinase Regulator), *IL1F10* (Interleukin 1 Family Member 10), and *IL1RN* (Interleukin 1 receptor antagonist). Recently, HNF4A has been known to have protective roles in inflammation via anti-angiogenesis, where the knockout of *HNF4A* promotes diabetic retinopathy [74]. The knockout of *ARNTL*, which is known to control circadian rhythms, has been shown to affect retinal development and accelerate cone photoreceptor degeneration during aging [75]. HNF1B primarily regulates the pathways involved in cell cycle progression and apoptosis. A defect in this gene causes maturity-onset diabetes in young people, as well as causing other cancers [76]. APCS or SAP (serum amyloid protein) is produced in the liver, is associated with inflammation, and has a characteristic pentameric organization like CRP. The removal of SAPs may be therapeutic in controlling inflammation [77]. IL1RN (interleukin 1 receptor antagonist) is known to have anti-inflammatory effects by inhibiting the activity of interleukin 1 (IL1) by binding to the IL-1 receptor [78]. 

Five genes are pro-inflammatory: *NECTIN2* (Nectin Cell Adhesion Molecule 2), *TOMM40* (Translocase of Outer Mitochondrial Membrane 40), *APOC4-APOC2* (Apolipoprotein C4 to C2), *APCS* (Amyloid P Component, Serum), and *NLRP3* (NLR Family Pyrin Domain Containing 3). Furthermore, when analyzing each protein’s expression via organ production, *APOC4-APOC2*, *GCKR*, *APCS*, and *SERPINA1* dominated in the liver, while *ARNTL* dominated in the brain. Other proteins were almost equally distributed in the whole body as seen via analysis with BioGPS [79]. From the literature reviews, we can see that *NECTIN2* has inflammatory roles in the brain [80], *TOMM40* causes neuro-inflammation and Alzheimer’s disease [81] and APOC4-APOC2 may affect the TOMM40-APOE-APOC2 axis, related to Alzheimer’s disease [82]. NLRP3 activates inflammasomes in response to membrane integrity abnormalities, resulting in the release of the inflammatory cytokine IL-1. This may lead to liver cirrhosis and brain diseases such as Alzheimer’s disease [83].

And, finally, one gene *IL6R* (interleukin 6 receptor) has both anti-inflammatory and pro-inflammatory roles. According to research by Ngwa et al., IL-6 activates STAT3 isoforms to both promote and control the production of CRP [84]. 

From the above analysis, we can see that an increased production of CRP by the liver and increasing CRP levels may cause damage via inflammatory recruitment processes; however, there are anti-inflammatory pathways that resolve further damage, especially in the brain, which is an immune sanctuary. Preliminary studies showing CRP mediating the liver–brain axis have been reported, especially with CRP in liver fibrosis and cognitive impairment [27]. Thus, CRP may be involved in pro-inflammatory and anti-inflammatory roles in the liver–brain–retina axis, as the retina is known to be an extension of the brain [28].

Our study’s primary strength is the comparatively large cohort dataset, which revealed a causal relationship between AMD and CRP. This study does, however, have many shortcomings. First, compared to Han et al., there is no significant difference except that the dataset is different when analyzing MR [42]. However, the significance of this research is two-fold: we have replicated similar results and have used two-sample MR; we used CRP from the BBJ project and UKB project and AMD from the IAMDGC and UKB data. Thus, there was a difference in IVs, as Han et al. had 44 IV SNPs, and our study had 204 IV SNPs. Second, we did not have access to individual-level data. Accordingly, the summary statistics based on two-sample MR could not account for the existence of multiple confounding variables. Third, the test methodologies employed to substantiate the MR assumptions were insufficient in providing comprehensive validation. Deviation from the MR assumptions might result in erroneous conclusions, thus warranting a cautious interpretation of the results. Fourth, few genome datasets included ophthalmic phenotype data; thus, it was difficult to separate and summarize a meta-analysis that included a portion of the UKB.

## 5. Conclusions

Our study demonstrated strong evidence of a potential causal association between CRP and AMD, with an increasing risk of AMD associated with elevated CRP levels in European and East Asian populations using MR analysis. Considering the significance of CRP levels, the association between CRP levels and AMD should be investigated.

## Figures and Tables

**Figure 1 biomedicines-12-00807-f001:**
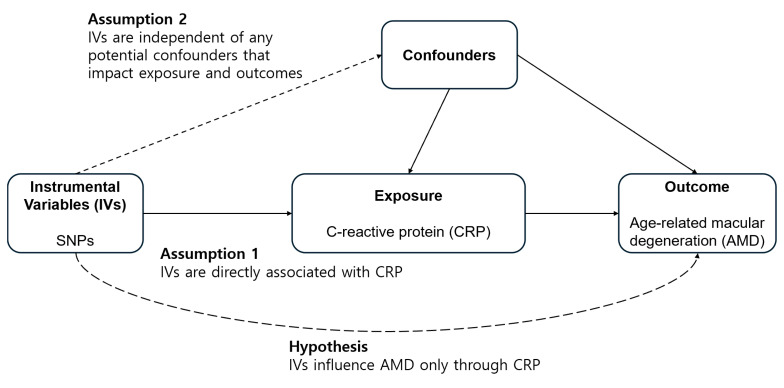
Illustration depicting a two-sample Mendelian randomization analysis. (SNP, single-nucleotide polymorphism).

**Figure 2 biomedicines-12-00807-f002:**
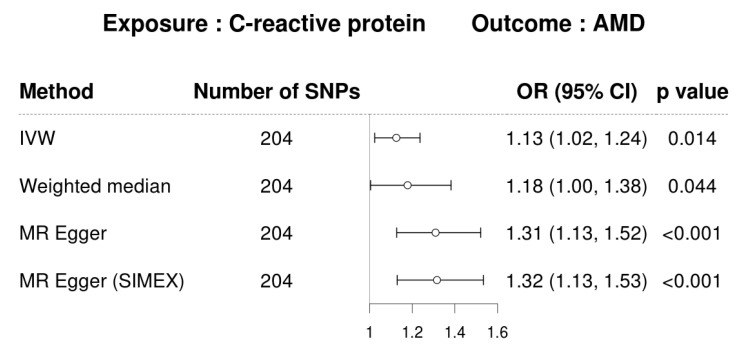
Forest plot of causal associations between CRP and AMD. (CRP, C-reactive protein; IVW, inverse-variance-weighted; SIMEX, simulation extrapolation; OR, odds ratio; CI, confidence interval).

**Figure 3 biomedicines-12-00807-f003:**
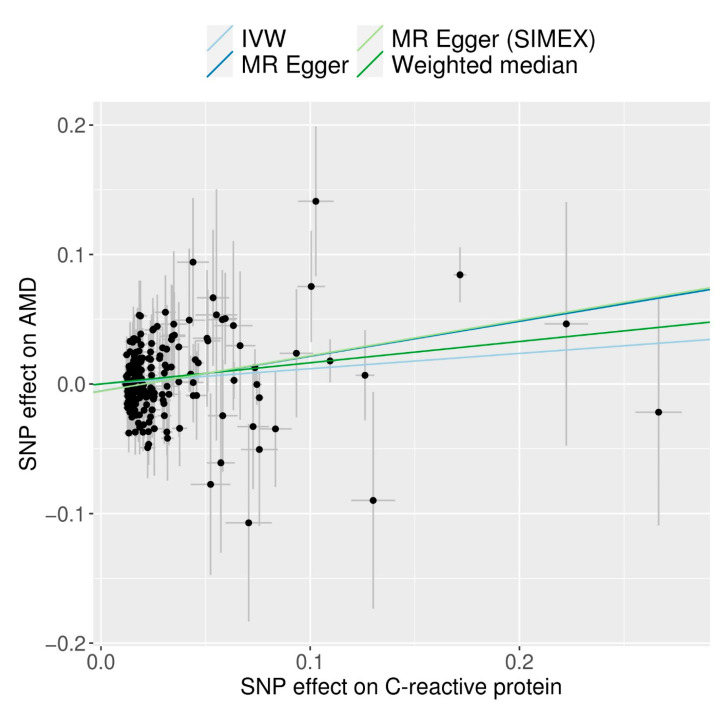
Scatter plots of MR tests assessing the effect of CRP levels on AMD (IVW, inverse-variance-weighted; MR, Mendelian randomization; SIMEX, simulation extrapolation).

**Figure 4 biomedicines-12-00807-f004:**
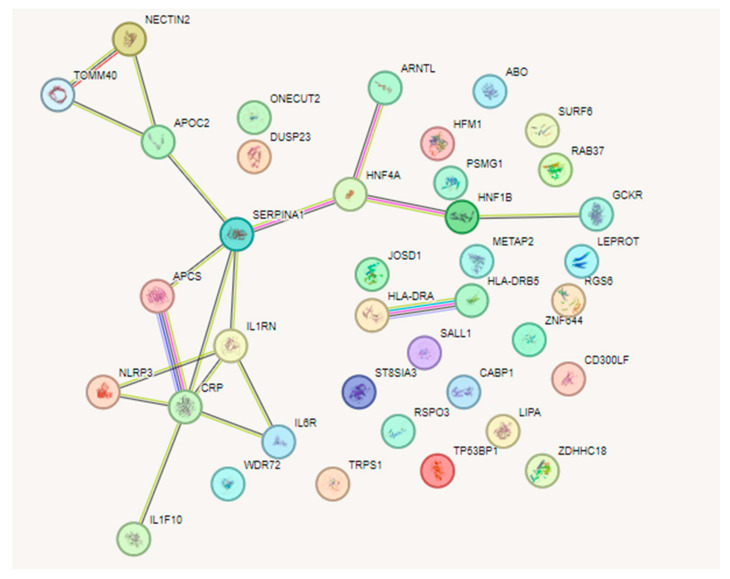
Network analysis with the selected IVs in Appendix A, via STRING database (https://string-db.org, accessed on 3 October 2023) [59]. F values over 100 were selected for analysis. The result is from the full-string network analysis simulation. PPI network shows 13 genes interacting with each other. The resulting PPI enrichment score *p*-value is 0.000323.

**Table 1 biomedicines-12-00807-t001:** Summary statistics of data sources.

Traits	Data Source	No. of Participants	Population	No. of Variants	Reference
CRP	BBJ Project + UKB	436,491	East Asian and European	20,529,698	[46]
AMD	Eleven sources of data including the IAMDGC and UKB	105,248 (14,034 cases + 91,214 controls)	European	11,703,383	[47]

BBJ, Biobank Japan; CRP, C-reactive protein; UKB, UK Biobank; AMD, age-related macular degeneration; IAMDGC, International AMD Genomics Consortium.

**Table 2 biomedicines-12-00807-t002:** Heterogeneity and horizontal pleiotropy of instrumental variables.

Exposure				Heterogeneity		Horizontal Pleiotropy				
				Cochran’s Q Test from IVW	Rücker’s Q Test from MR-Egger	MR-PRESSO Global Test	MR-Egger		MR-Egger (SIMEX)	
	**N**	**F**	**I^2^ (%)**	***p*-Value**	***p*-Value**	***p*-Value**	**Intercept, β (SE)**	***p*-Value**	**Intercept, β (SE)**	***p*-Value**
C-reactive protein	204	113.47	97.45	0.042	0.074	0.032	−0.006 (0.002)	0.012	−0.006 (0.002)	0.012

N, number of instruments; F, mean F statistic; IVW, inverse-variance-weighted; MR, Mendelian randomization; PRESSO, pleiotropy sum of residuals and outliers; SIMEX, simulation extrapolation; β, beta coefficient; SE, standard error.

**Table 3 biomedicines-12-00807-t003:** Selected set of single-nucleotide polymorphisms as instrumental variables that influence AMD only through CRP and have associations with CRP.

SNP	Chr	Position	Nearby Gene	Effect Allele	Other Allele	Effect Allele Frequency	β	SE	*p*-Value	F
rs12972970	19	45387596	*NECTIN2*	A	G	0.152	−0.172	0.0030	1.00 × 10^−200^	3271.84
rs117310449	19	45393516	*TOMM40*	T	C	0.012	−0.267	0.0111	2.74 × 10^−128^	576.43
rs10420434	19	45451190	*APOC4-APOC2*	A	G	0.047	0.059	0.0057	1.65 × 10^−25^	108.60
rs28929474	14	94844947	*SERPINA1*	T	C	0.020	−0.103	0.0085	2.27 × 10^−33^	145.98
rs1800961	20	43042364	*HNF4A*	T	C	0.027	−0.101	0.0066	3.55 × 10^−52^	231.87
rs1037169	11	13361005	*ARNTL*	C	T	0.688	0.029	0.0026	2.68 × 10^−29^	127.00
rs17138478	17	36073320	*HNF1B*	A	C	0.144	0.032	0.0030	2.14 × 10^−25^	112.36
rs1260326	2	27730940	*GCKR*	C	T	0.575	−0.064	0.0022	3.85 × 10^−187^	835.74
rs1811471	1	159642599	*APCS*	A	G	0.258	0.110	0.0028	1.00 × 10^−200^	1529.37
rs55709272	2	113867288	*IL1F10; IL1RN*	C	T	0.364	0.043	0.0024	2.27 × 10^−73^	319.52
rs4133213	1	154395212	*IL6R*	A	C	0.442	−0.076	0.0022	1.00 × 10^−200^	1183.99
rs56015600	1	247601886	*NLRP3*	G	A	0.622	0.034	0.0022	4.09 × 10^−52^	233.26

SNP, single-nucleotide polymorphism; Chr, chromosome; β, beta coefficient; SE, standard error; F, mean F statistic; AMD, age-related macular degeneration.

## Data Availability

The datasets used and/or analyzed in the current study are available from Biobank Japan (BBJ https://pheweb.jp/, accessed on 7 March 2023) [46] and the GWAS catalogue (https://www.ebi.ac.uk/gwas/summary-statistics, accessed on 19 July 2022).

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
