# Peer review of "Potential Causal Association between C-Reactive Protein Levels in Age-Related Macular Degeneration: A Two-Sample Mendelian Randomization Study"

_biomedicines, 2024, doi:10.3390/biomedicines12040807_

Round 1

Reviewer 1 Report

Comments and Suggestions for Authors

The authors investigated the causal effects of CRP levels on AMD by two-sample Mendelian Randomization analyze. Here, several issues need to be clarified:

1. There have been similar studies evaluating the effect of CRP levels on AMD (PMID: 31900758), and this one by the authors is not significantly innovative compared to them. In addition, the research method of this paper lacks novelty. 

2. The article only focused on CRP and AMD, without considering several types of AMD.

3. The authors focused on the relationship between rs4133213 and rs55709272 SNP loci and inflammation, without clearly identifying the relevant value of other SNP loci.

4. Figure 4 only provides a schematic diagram of the PPI network analysis, without F-values and beta values of APCS, IL1RN, NLRP3, IL6R, and ILF10 in graphics or tables. The F-values and beta values of IL6R related SNPs were not mentioned as well. In addition, functional enrichment analysis is important in PPI network analysis, which is not shown in the results and discussion.

5. In discussion section, the author explained the negative beta values of IL6R related SNPs as negative feedback system, which was lack of result or literature support.

Author Response

Reviewer 1

Comments and Suggestions for Authors

The authors investigated the causal effects of CRP levels on AMD by two-sample Mendelian Randomization analyze. Here, several issues need to be clarified:

  1. There have been similar studies evaluating the effect of CRP levels on AMD (PMID: 31900758), and this one by the authors is not significantly innovative compared to them. In addition, the research method of this paper lacks novelty.

→ Thank you for critical reviews. The authors are aware of similar studies and are concerned about what differentiates them. Additionally, because the data sets used for analysis are different and the number of subjects is large, we think the research is worthwhile even if creativity is low. While existing studies showed Mendelian randomization results for CRP and AMD, this study analyzed candidate genes in addition to those. This is because existing research has new aspects to show concerns about why there is a causal connection with CRP through the liver-brain-retina axis. We added these comments in the discussion section in lines 223-239.

Also, we believe that although this may have less innovation compared to PMID 31900758, we have also replicated similar results using several sets of data, CRP from the BJJ project and UKB project, AMD from the IAMDGC and UKB data in lines 211-222. The PMID 31900758 had 44 SNPs as instrumental variables, whereas our study had 204.

We have added this in lines 310-315.

First, compared to Han et al., there is no significant difference except that the dataset is different when analyzing MR [42]. However, the significance of this research is two-fold: we have replicated similar results, and have used two-sample MR, where we used CRP from the BJJ project and UKB project, and AMD from the IAMDGC and UKB data. Thus, there was a difference of IVs where Han et al. had 44 IV SNPs, and our study had 204 IV SNPs.

  1. The article only focused on CRP and AMD, without considering several types of AMD.

→ Thank you for your comments. AMD is largely divided into early and late, and previous studies (PMID: 31900758) have shown causality, so we understood it from a different perspective than showing subtypes with the same data. One of the starting points was from our research PMID: 33618707, where we calculated polygenetic risk score to predict AMD between ethnicities for any AMD and late AMD. Although this model fits well in the early and late AMD models for the age lower than 45 years old group, it does not fit well for ages above 65 years. Thus, we tried to analyze if there were any other factors such as CRP or confounding factors. This is explained in lines 281-291.

  1. The authors focused on the relationship between rs4133213 and rs55709272 SNP loci and inflammation, without clearly identifying the relevant value of other SNP loci.

→ Thank you for your comments. We have added this in the discussion of lines 292-331.

Some considerations can be made based on the network analysis in Figure 4. of the resulting SNPs as IV. There are 13 genes associated with CRP as in Table 3. Seven genes are anti-inflammatory in the increase of expression: SERPINA1 (Serpin Family A Member 1), HNF4A (Hepatocyte Nuclear Factor 4 Alpha), ARNTL (aryl hydrocarbon receptor nu-clear translocator-like protein 1), HNF1B (HNF1 Homeobox B), GCKR (Glucokinase Regulator), IL1F10 (Interleukin 1 Family Member 10), IL1RN (Interleukin 1 receptor antagonist). Recently, HNF4A has been known to have protective roles in inflammation by anti-angiogenesis, where knockout of HNF4A promotes diabetic retinopathy [74]. ARNTL which is known to control circadian rhythms, has known that knockout affects retinal development and accelerates cone photoreceptor degeneration during aging [75]. HNF1B performs its function mainly by regulating the cell cycle and apoptosis pathways. A defect of this gene causes maturity-onset diabetes in the young and other cancers [76]. APCS or SAP (Serum Amyloid Protein) is produced in the liver and is associated with inflammation, which has a characteristic pentameric organization like CRP. Removal of SAPs may be therapeutic in controlling inflammation [77]. IL1RN (Interleukin 1 receptor antagonist) is known to have anti-inflammatory effects, by inhibiting the activity of interleukin 1 (IL1) by binding to the IL-1 receptor [78].

Five genes are pro-inflammatory: NECTIN2 (Nectin Cell Adhesion Molecule 2), TOMM40 (Translocase of Outer Mitochondrial Membrane 40), APOC4-APOC2 (Apolipoprotein C4 to C2), APCS (Amyloid P Component, Serum), NLRP3 (NLR Family Pyrin Do-main Containing 3). Furthermore, when analyzing each protein expression via organ production, APOC4-APOC2, GCKR, APCS, SERPINA1 dominated in the liver, while ARNTL dominated in the brain. Other proteins were almost equally distributed in the whole body via analysis with BioGPS [79]. From the literature reviews, NECTIN2 has inflammatory roles in the brain [80], TOMM40 with neuroinflammation and Alzheimer’s disease [81], and APOC4-APOC2 may have TOMM40-APOE-APOC2 axis, related to Alzheimer’s disease [82]. NLRP3 mediates inflammasome activation in response to defects in membrane integrity, leading to the secretion of inflammatory cytokines IL-1. This may lead to liver cirrhosis and brain diseases such as Alzheimer’s disease [83].

And finally, one gene IL6R (Interleukin 6 Receptor) has both anti-inflammatory and pro-inflammatory roles. Ngwa et al. reported that IL-6 not only induces CRP expression but also regulates the in-duction of CRP expression by activating STAT3 isoforms [84].

From the above analysis, we can see that increasing production of CRP from the liver and increasing CRP levels may cause damage via inflammatory recruitment processes, however, there exists anti-inflammatory pathways which resolve further damage, espe-cially in the brain, which is an immune sanctuary. Preliminary studies of CRP mediating the liver–brain axis have been reported, especially with CRP in liver fibrosis and cognitive impairment [27]. Thus, CRP may be involved in pro-inflammatory and anti-inflammatory roles in the liver–brain–retina axis, as the retina is known to be the window of the brain [28].

  1. Figure 4 only provides a schematic diagram of the PPI network analysis, without F-values and beta values of APCS, IL1RN, NLRP3, IL6R, and ILF10 in graphics or tables. The F-values and beta values of IL6R related SNPs were not mentioned as well. In addition, functional enrichment analysis is important in PPI network analysis, which is not shown in the results and discussion.

→ Thank you for your comments. We added the following in the result (lines 200-201) and Figure 4 (lines 208-209).

The PPI network showed 16 genes interacting with each other, with the PPI enrichment score p-value as 0.000323.

  1. In discussion section, the author explained the negative beta values of IL6R related SNPs as negative feedback system, which was lack of result or literature support.

→ Thank you for your comments. We have added this in the discussion of lines 292-331.

Some considerations can be made based on the network analysis in Figure 4. of the resulting SNPs as IV. There are 13 genes associated with CRP as in Table 3. Seven genes are anti-inflammatory in the increase of expression: SERPINA1 (Serpin Family A Member 1), HNF4A (Hepatocyte Nuclear Factor 4 Alpha), ARNTL (aryl hydrocarbon receptor nu-clear translocator-like protein 1), HNF1B (HNF1 Homeobox B), GCKR (Glucokinase Regu-lator), IL1F10 (Interleukin 1 Family Member 10), IL1RN (Interleukin 1 receptor antagonist). Recently, HNF4A has been known to have protective roles in inflammation by antiangio-genesis, where knockout of HNF4A promoting diabetic retinopathy [74]. ARNTL which is known to control circadian rhythms, has known that knockout affects retinal development and accelerates cone photoreceptor degeneration during aging [75]. HNF1B performs its function mainly by regulating the cell cycle and apoptosis pathways. A defect of this gene causes maturity-onset diabetes in the young and other cancers [76]. APCS or SAP (Serum Amyloid Protein) is produced in the liver and is associated with inflammation, which has a characteristic pentameric organization like CRP. Removal of SAPs may be therapeutic in controlling inflammation [77]. IL1RN (Interleukin 1 receptor antagonist) is known to have anti-inflammatory effects, by inhibiting the activity of interleukin 1 (IL1) by binding to the IL-1 receptor [78].

Five genes are pro-inflammatory: NECTIN2 (Nectin Cell Adhesion Molecule 2), TOMM40 (Translocase of Outer Mitochondrial Membrane 40), APOC4-APOC2 (Apolipo-protein C4 to C2), APCS (Amyloid P Component, Serum), NLRP3 (NLR Family Pyrin Do-main Containing 3). Furthermore, when analyzing each protein expression via organ production, APOC4-APOC2, GCKR, APCS, SERPINA1 dominated in the liver, while ARNTL dominated in the brain. Other proteins were almost equally distributed in the whole body via analysis with BioGPS [79]. From the literature reviews, NECTIN2 has inflammatory roles in the brain [80], TOMM40 with neuroinflammation and Alzheimer’s disease [81] and APOC4-APOC2 may have TOMM40-APOE-APOC2 axis, related to Alzheimer’s dis-ease [82]. NLRP3 mediates inflammasome activation in response to defects in membrane integrity, leading to the secretion of inflammatory cytokines IL-1. This may lead to liver cirrhosis and brain diseases such as Alzheimer’s disease [83].

And finally, one gene IL6R (Interleukin 6 Receptor) has both anti-inflammatory and pro-inflammatory roles. Ngwa et al. reported that IL-6 not only induces CRP expression but also regulates the induction of CRP expression by activating STAT3 isoforms [84].

From the above analysis, we can see that increasing production of CRP from the liver and increasing CRP levels may cause damage via inflammatory recruitment processes, however, there exists anti-inflammatory pathways which resolve further damage, especially in the brain, which is an immune sanctuary. Preliminary studies of CRP mediating the liver–brain axis have been reported, especially with CRP in liver fibrosis and cognitive impairment [27]. Thus, CRP may be involved in pro-inflammatory and anti-inflammatory roles in the liver–brain–retina axis, as the retina is known to be the window of the brain [28].

Reviewer 2 Report

Comments and Suggestions for Authors

This study is significant as it aims to determine the causal relationship between serum CRP and AMD. Previous attempts have been made to understand this association; however, this meta-analytic study stands out by incorporating a more extensive cohort GWAS dataset and conducting more precise statistical analysis. This approach potentially offers more robust and reliable insights into the relationship between CRP levels and the risk of AMD. 

The authors used a dataset from a GWAS-based meta-analysis of BBJ and UKB instead of solely UKB. The network analysis revealed that the IV SNPs with positive beta values associated with CRP are linked to higher proximity, while those with negative beta values are not. Consequently, their conclusion is that they have demonstrated an increased risk of AMD associated with elevated CRP levels. Some concerns are described below.

The SNP data for AMD outcomes appear to be from references studying European populations. If so, is this clearly stated in the manuscript?

Regarding the IVs shown in Figure 4, how many of these SNPs are associated with AMD? If they are not directly associated with the onset of AMD, is CRP supposed to increase independently of complement-related genotypes?

Although the SNPs in IL6R (rs4133213) and IL1RN (rs55709272) are discussed, a comprehensive analysis should be made for SNPs associated with CRP to interpret the roles of increased CRP and SNPs in AMD.

Author Response

Comments and Suggestions for Authors

This study is significant as it aims to determine the causal relationship between serum CRP and AMD. Previous attempts have been made to understand this association; however, this meta-analytic study stands out by incorporating a more extensive cohort GWAS dataset and conducting more precise statistical analysis. This approach potentially offers more robust and reliable insights into the relationship between CRP levels and the risk of AMD.

→ Thank you very much for your kind comment.

The authors used a dataset from a GWAS-based meta-analysis of BBJ and UKB instead of solely UKB. The network analysis revealed that the IV SNPs with positive beta values associated with CRP are linked to higher proximity, while those with negative beta values are not. Consequently, their conclusion is that they have demonstrated an increased risk of AMD associated with elevated CRP levels. Some concerns are described below.

 The SNP data for AMD outcomes appear to be from references studying European populations. If so, is this clearly stated in the manuscript?

→ Thank you very much for this comment. We have modified the manuscript to include this in lines 232-233:

Second, we used the summary statistics of 11 sources of IAMDGC GWAS data for the outcome data, consisting of SNPs from European descendants.

Regarding the IVs shown in Figure 4, how many of these SNPs are associated with AMD? If they are not directly associated with the onset of AMD, is CRP supposed to increase independently of complement-related genotypes? 

→ Thank you very much for this comment. We included this in Table 3. All of these genes are not associated directly with AMD, but through CRP. We have explained this in lines 203-204.

F values over 10 signify that these genes are not associated directly with AMD but through CRP.

Although the SNPs in IL6R (rs4133213) and IL1RN (rs55709272) are discussed, a comprehensive analysis should be made for SNPs associated with CRP to interpret the roles of increased CRP and SNPs in AMD.

→ Thank you for your comments. We have added this in the discussion of lines 292-331.

Some considerations can be made based on the network analysis in Figure 4. of the resulting SNPs as IV. There are 13 genes associated with CRP as in Table 3. Seven genes are anti-inflammatory in the increase of expression: SERPINA1 (Serpin Family A Member 1), HNF4A (Hepatocyte Nuclear Factor 4 Alpha), ARNTL (aryl hydrocarbon receptor nu-clear translocator-like protein 1), HNF1B (HNF1 Homeobox B), GCKR (Glucokinase Regu-lator), IL1F10 (Interleukin 1 Family Member 10), IL1RN (Interleukin 1 receptor antagonist). Recently, HNF4A has been known to have protective roles in inflammation by antiangio-genesis, where knockout of HNF4A promoting diabetic retinopathy [74]. ARNTL which is known to control circadian rhythms, has known that knockout affects retinal development and accelerates cone photoreceptor degeneration during aging [75]. HNF1B performs its function mainly by regulating the cell cycle and apoptosis pathways. A defect of this gene causes maturity-onset diabetes in the young and other cancers [76]. APCS or SAP (Serum Amyloid Protein) is produced in the liver and is associated with inflammation, which has a characteristic pentameric organization like CRP. Removal of SAPs may be therapeutic in controlling inflammation [77]. IL1RN (Interleukin 1 receptor antagonist) is known to have anti-inflammatory effects, by inhibiting the activity of interleukin 1 (IL1) by binding to the IL-1 receptor [78].

Five genes are pro-inflammatory: NECTIN2 (Nectin Cell Adhesion Molecule 2), TOMM40 (Translocase of Outer Mitochondrial Membrane 40), APOC4-APOC2 (Apolipo-protein C4 to C2), APCS (Amyloid P Component, Serum), NLRP3 (NLR Family Pyrin Do-main Containing 3). Furthermore, when analyzing each protein expression via organ production, APOC4-APOC2, GCKR, APCS, SERPINA1 dominated in the liver, while ARNTL dominated in the brain. Other proteins were almost equally distributed in the whole body via analysis with BioGPS [79]. From the literature reviews, NECTIN2 has inflammatory roles in the brain [80], TOMM40 with neuroinflammation and Alzheimer’s disease [81] and APOC4-APOC2 may have TOMM40-APOE-APOC2 axis, related to Alzheimer’s dis-ease [82]. NLRP3 mediates inflammasome activation in response to defects in membrane integrity, leading to the secretion of inflammatory cytokines IL-1. This may lead to liver cirrhosis and brain diseases such as Alzheimer’s disease [83].

And finally, one gene IL6R (Interleukin 6 Receptor) has both anti-inflammatory and pro-inflammatory roles. Ngwa et al. reported that IL-6 not only induces CRP expression but also regulates the induction of CRP expression by activating STAT3 isoforms [84].

From the above analysis, we can see that increasing production of CRP from the liver and increasing CRP levels may cause damage via inflammatory recruitment processes, however, there exists anti-inflammatory pathways which resolve further damage, especially in the brain, which is an immune sanctuary. Preliminary studies of CRP mediating the liver–brain axis have been reported, especially with CRP in liver fibrosis and cognitive impairment [27]. Thus, CRP may be involved in pro-inflammatory and anti-inflammatory roles in the liver–brain–retina axis, as the retina is known to be the window of the brain [28].

Reviewer 3 Report

Comments and Suggestions for Authors

The workpaper is well written and formulated. I suggest improving Figure 1 and including Table S1 not as supplementary but in the text.

Author Response

Comments and Suggestions for Authors

The workpaper is well written and formulated. I suggest improving Figure 1 and including Table S1 not as supplementary but in the text.

→ Thank you very much. We have modified Figure 1 and summarized eleven IVs related to CRP to AMD in Table 3 from Supplemental Table S1. The list in Supplemental Table S1 consists of 204 genes, so we selected 14 genes directly related to CRP in Figure 4. This is added in the result section of lines 201-204.

IVs that influence AMD only through CRP that are associated with each other are in Table 3, with an F-value of over 100. F values over 10 signify that these genes are not associated directly with AMD but through CRP.

Reviewer 4 Report

Comments and Suggestions for Authors

Manuscript «Сausal association between С-reactive protein levels in age- related macular degeneration: a two-sample mendelian randomization study» of Woo et al. is devoted to research of causal relationship between CRP and AMD. CRP is a biomarker of inflammation. It can act as pro-inflammatory or anti-inflammatory molecule, depending on form. Native CRP exhibit anti-inflammatory role, monomeric form promoted chemotaxis of leukocytes. However, observational studies have shown mixed conclusions regarding the association between circulating CRP levels and the risk of AMD. There was previously one Mendelian randomization study that showed that higher CRP levels increase the risk for all forms of AMD. The authors of manuscript conducted a new Mendelian randomization study using different approach, since statistical techniques of MR are constantly evolving. In addition, the results of the MR analysis may vary based on the selection of proxies for CRP, and large datasets combining the meta-analyses of Biobank Japan and UK Biobank  are expected to generate more substantial results. Thus, the authors investigated the causal effects of CRP on the risk of AMD using multi-ethnic population (n=436,491 for CRP)  from 2 databases. Methods: the inverse-variance weighted method, weighted median method, MR-–Egger method, and MR–PRESSO.

In results the authors identified 204 IVs that had a significant association with CRP. Based on MR analysis CRP demonstrated a significant causal association with AMD using all 4 methods with modest OR approx. 1.2.

The limitations of study are acknowledged by the authors: « First, we did not have access to individual-level data. Thus, we were unable  to explain the presence of numerous confounding factors using summary statistics based on two-sample MR. Second, the test procedures used to validate the MR hypotheses did not provide complete validation. Violations of MR assumptions can lead to invalid conclusions, THUS WARRANTING A CAUTIOUS INTERPRETATION OF THE RESULTS». In this regard, I suggest that the authors remove the word «causal» from the title, especially since the method is indicated in the title, which implies a potential causal relationship. Or add the word Potential in front, softening the interpretation.

Minor comment:

 Line 21 and 140-141 MR–PRESSO MR-–Pleiotropy Residual Sum and Outlier OR? polyhedral sum of residuals and outliers. I think it is necessary to use a uniform version of the decryption of this abbreviation

Author Response

Manuscript «Сausal association between С-reactive protein levels in age- related macular degeneration: a two-sample mendelian randomization study» of Woo et al. is devoted to research of causal relationship between CRP and AMD. CRP is a biomarker of inflammation. It can act as pro-inflammatory or anti-inflammatory molecule, depending on form. Native CRP exhibit anti-inflammatory role, monomeric form promoted chemotaxis of leukocytes. However, observational studies have shown mixed conclusions regarding the association between circulating CRP levels and the risk of AMD. There was previously one Mendelian randomization study that showed that higher CRP levels increase the risk for all forms of AMD. The authors of manuscript conducted a new Mendelian randomization study using different approach, since statistical techniques of MR are constantly evolving. In addition, the results of the MR analysis may vary based on the selection of proxies for CRP, and large datasets combining the meta-analyses of Biobank Japan and UK Biobank are expected to generate more substantial results. Thus, the authors investigated the causal effects of CRP on the risk of AMD using multi-ethnic population (n=436,491 for CRP) from 2 databases. Methods: the inverse-variance weighted method, weighted median method, MR-–Egger method, and MR–PRESSO.

In results the authors identified 204 IVs that had a significant association with CRP. Based on MR analysis CRP demonstrated a significant causal association with AMD using all 4 methods with modest OR approx. 1.2.

The limitations of study are acknowledged by the authors: « First, we did not have access to individual-level data. Thus, we were unable  to explain the presence of numerous confounding factors using summary statistics based on two-sample MR. Second, the test procedures used to validate the MR hypotheses did not provide complete validation. Violations of MR assumptions can lead to invalid conclusions, THUS WARRANTING A CAUTIOUS INTERPRETATION OF THE RESULTS». In this regard, I suggest that the authors remove the word «causal» from the title, especially since the method is indicated in the title, which implies a potential causal relationship. Or add the word Potential in front, softening the interpretation.

 → Thank you very much. We have added Potential in front of the title as suggested. Also in the conclusion, we have modified this in line 347.

Line 308: Our study demonstrated strong evidence of a potential causal association between CRP and AMD

Minor comment:

 Line 21 and 140-141 MR–PRESSO MR-–Pleiotropy Residual Sum and Outlier OR? polyhedral sum of residuals and outliers. I think it is necessary to use a uniform version of the decryption of this abbreviation 

→ Thank you very much. We have changed polyhedral to pleiotropy in lines 140-141 and the footnote of Table 2.

Line 140-141: and the MR pleiotropy sum of residuals and outliers (MR–PRESSO)

Footnote of Table 2: N, number of instruments; F, mean F statistic; IVW, inverse-variance weighed; MR, Mendelian randomization; PRESSO, pleiotropy sum of residuals and outliers; SIMEX, simulation extrapolation; β, beta coefficient; SE, standard error;

Round 2

Reviewer 1 Report

Comments and Suggestions for Authors

In their revised manuscript, the authors have adequately responded to the comments provided in the original review’s. They also included more discussions for helping comprehensively explain the current challenges and future directions based on the content. Although with less innovation, this could be a supplement to the field.

Author Response

Thank you for the opportunity to submit this paper to Biomedicines, and we also thank you for the great critical reviews.

Here are the responses for reviewer 1.

Dear In their revised manuscript, the authors have adequately responded to the comments provided in the original review’s. They also included more discussions for helping comprehensively explain the current challenges and future directions based on the content. Although with less innovation, this could be a supplement to the field.

→ Thank you very much for this comment.